# Sound and Complete Neurosymbolic Reasoning with LLM-Grounded Interpretations

**Bradley P. Allen**                                              B.P.ALLEN@UVA.NL
*University of Amsterdam, Amsterdam, NL*

**Prateek Chhikara**                              PRATEEKCHHIKARA24@GMAIL.COM
*University of Southern California, CA, US*

**Thomas Macaulay Ferguson**                              FERGUT5@RPI.EDU
*Rensselaer Polytechnic Institute, Troy, NY, US*

**Filip Ilievski**                                              F.ILIEVSKI@VU.NL
*Vrije Universiteit Amsterdam, Amsterdam, NL*

**Paul Groth**                                              P.T.GROTH@UVA.NL
*University of Amsterdam, Amsterdam, NL*

**Editors:** Leilani H. Gilpin, Eleonora Giunchiglia, Pascal Hitzler, and Emile van Krieken

## Abstract

Large language models (LLMs) have demonstrated impressive capabilities in natural language understanding and generation, but they exhibit problems with logical consistency in the output they generate. How can we harness LLMs' broad-coverage parametric knowledge in formal reasoning despite their inconsistency? We present a method for directly integrating an LLM into the interpretation function of the formal semantics for a paraconsistent logic. We provide experimental evidence for the feasibility of the method by evaluating the function using datasets created from several short-form factuality benchmarks. Unlike prior work, our method offers a theoretical framework for neurosymbolic reasoning that leverages an LLM's knowledge while preserving the underlying logic's soundness and completeness properties.

## 1. Introduction

Applications involving commonsense reasoning and biomedical knowledge, where inconsistencies and incomplete information are commonplace, remain challenging frontiers for AI systems. While LLMs encode vast parametric knowledge (Petroni et al., 2019), they also suffer from inconsistency and incompleteness (Cheng et al., 2025), limiting their use as knowledge bases for such applications. Efforts to connect logical reasoning with LLMs relying on prompting strategies and external symbolic solvers (Wei et al., 2022; Cheng et al., 2025) show potential but exhibit significant shortcomings as well Hoppe et al. (2025), lacking formal frameworks for managing LLM knowledge inconsistency and incompleteness. *Paraconsistent logics* (Priest et al., 2025) are multi-valued non-classical logics that handle inconsistent information without logical explosion, where contradictions would make everything provable. *Belnap computers* (Belnap, 1977a,b) are theoretical constructions described by Nuel Belnap to model contexts in which machines are responsible for reasoning in the face of incomplete or inconsistent information. This raises a natural question: can paraconsistent logic and Belnap computers provide a more natural framework for using LLMs as knowledge bases, despite their inconsistency and incompleteness?

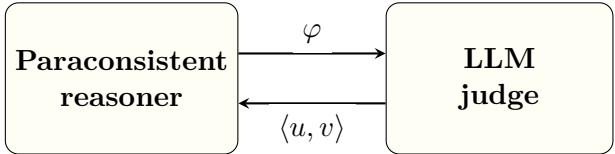

Figure 1: A Belnap computer using an LLM judge as a source of knowledge. Let $\mathcal{L}$ be the object language for a paraconsistent logic, and let $\mathcal{L}_{AT}$ be the set of atomic formulas. A paraconsistent reasoner (left) sends an atomic formula $\varphi \in \mathcal{L}_{AT}$ to the LLM judge (right), which returns a generalized truth value $\langle u, v \rangle$, such that $u$ indicates if the LLM judge was able to verify $\varphi$, and $v$ indicates if the LLM judge was able to refute $\varphi$.

We propose a Belnap computer using an *LLM judge* as an external knowledge source. LLM judges scale factuality evaluation of LLM output for short- or long-form question answering tasks, returning truth valuations for statements in the LLM's output (Li et al., 2024). In our approach, an LLM judge responds to a query for the valuation of an atomic formula in the context of a paraconsistent reasoner with a *generalized truth value* (Shramko and Wansing, 2025, 2011). Generalized truth values allow the LLM judge to provide information to the reasoner not only about the degree of truth of an atomic formula given an LLM's parametric knowledge, but also with respect to the degree of knowledge the LLM has about the formula.

Section 2 discusses related work, and Section 3 defines ***(i) a bilateral factuality evaluation function*** that provides information beyond current LLM judge approaches to factuality evaluation. Section 4 integrates the function directly into the formal semantics of a paraconsistent logic through ***(ii) an LLM-grounded interpretation*** that preserves the soundness and completeness of analytic tableau systems for reasoning in the logic. Section 5 provides ***(iii) empirical evidence for practical implementation***, presenting evaluation findings using benchmarks derived from short-form factuality benchmarks and discussing limitations. Section 6 concludes with a summary and discussion of future work.

## 2. Related work

**Logical reasoning with LLMs**   Current approaches to reasoning with LLMs (Hoppe et al., 2025; Cheng et al., 2025) appear in Figure 2. In a *prompt-based* approach ($a$), an LLM is prompted with a verbalization $\delta(\Gamma) \in \Sigma^*$ of a set of formulas $\Gamma$, and performs natural language reasoning to produce a verbalization $\delta(\varphi)$ of $\varphi$ (Wei et al., 2022; Kojima et al., 2022; Dhuliawala et al., 2023; Yao et al., 2023). In a *solver-based* approach ($b$), an LLM is prompted with a verbalization of a set of formulas and produces a set of formulas $\Gamma$ in an object language $\mathcal{L}$, which a reasoner uses to deduce $\varphi$ (Pan et al., 2023; Olausson et al., 2023; Callewaert et al., 2025). In an approach based on *pre-training or fine-tuning* ($c$), a reasoner provides a training set of proofs $\Pi$, and the LLM learns from that to reason as in ($a$) (Jiao et al., 2023; Morishita et al., 2024; Feng et al., 2024; Liu et al., 2025). Unlike approaches ($b$) and ($c$) that use LLMs alongside reasoning systems, in our proposed *interpretation-based* approach ($d$) we integrate LLMs directly into the formal semantics of the reasoner's logic itself, by using an LLM to implement an interpretation function $\mathcal{I}$ to be used by the reasoner in inferring $\varphi$. This allows us to provide formal guarantees about

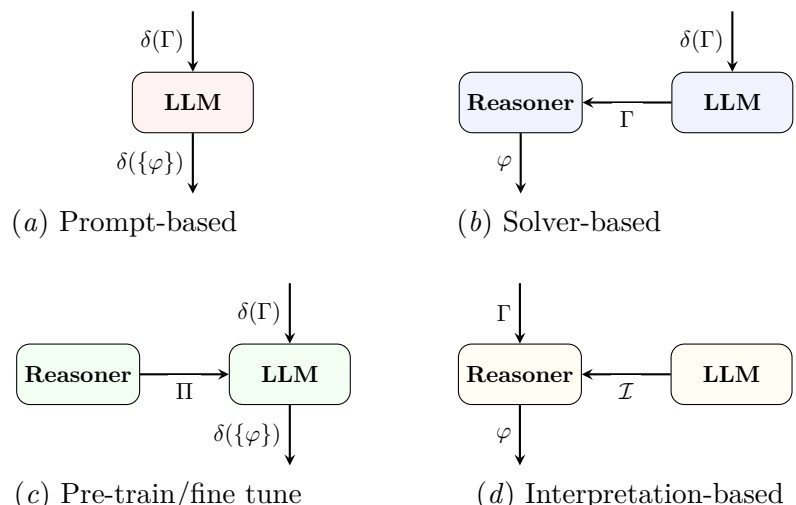

Figure 2: Four different approaches to logical reasoning with LLMs. Let $\mathcal{L}$ be a first-order language, $\Gamma$ be a set of statements in $\mathcal{L}$, $\varphi$ be a statement in $\mathcal{L}$, $\mathcal{I}$ be an interpretation for $\mathcal{L}$, $\Pi$ be a set of proofs of statements in $\mathcal{L}$, and $\delta : \mathcal{P}(\mathcal{L}) \to \Sigma^*$ be a verbalization function that takes a set of formulas in $\mathcal{L}$ and returns a natural language translation of the formulas. In each approach, we show how reasoning is performed in the context of generating a formal or natural language proof showing that $\Gamma \vdash \varphi$.

the soundness and completeness of the reasoning process that incorporates the LLM. In contrast with approaches ($a$), ($b$), and ($c$), instead of trying to get an LLM to reason using logic, we get a logic to reason using an LLM.

**Factuality evaluation using LLM judges** Factuality evaluation is the assessment of whether the output of a language model is factually correct (Wang et al., 2023; Bang et al., 2025). Recent work has focused on LLM judges (Zheng et al., 2023; Zhu et al., 2023) as a means to scale factuality evaluation, by prompting an LLM to produce a truth value assignment of the output of an LLM that is being evaluated, specifically a short- (Wei et al., 2024) or long-form (Jacovi et al., 2025) answer to a question. We apply LLM judges to assign generalized truth values to natural language translations of atomic formulas, taking a single-answer grading approach (Zheng et al., 2023). In contrast with current approaches to factuality evaluation, the use of generalized truth values provides information as to the LLM's epistemic stance towards the formula in question.

**Multi-valued logics for paraconsistent reasoning** Work building on Belnap's four-valued semantics has led to the development of a range of non-classical paraconsistent logics. Patel-Schneider (1989) showed how this idea could be applied to make terminological logics capable of performing subsumption correctly in the presence of contradictory knowledge. Kamide (2010), Ma et al. (2007), and Maier et al. (2013) expanded on this work to define a number of paraconsistent description logics. More recently, Ferguson (2017b) has proposed a computational interpretation of versions of first degree entailment (**FDE**) and Richard Angell's logic of analytic containment (**AC**). This interpretation models reasoners using

these logics as Belnap computers, and leads to a formal framework for **FDE** and **AC** as bilateral logics with sound and complete analytic tableau systems. We show how an LLM judge can be used to provide an interpretation for **AC** that preserves the soundness and completeness of the tableau system **ACrQ**, with applications to description logics as described in (Ferguson, 2021a).

## 3. Bilateral factuality evaluation of an atomic formula using an LLM

**AC** is a *conceptivist* logic (Ferguson, 2017a) that addresses hierarchical relationships between concepts. While in this work we focus on **AC** as a paraconsistent logic, it is also *paracomplete*, i.e., it rejects the law of excluded middle. The combination of those two properties makes it particularly suitable for applications involving vague predicates, incomplete information, or situations where classical logic's demands for both consistency and completeness are too strong. Appendix A provides a definition of **AC** with restricted quantification, which is necessary to support concept subsumption and existential quantification of roles when used as a description logic (Ferguson, 2021b). This definition treats **AC** as a *bilateral* logic, i.e., a logic which manages values for both the truth and falsity of a formula separately. Bilateralism in philosophical logic (Rumfitt, 2000) holds that understanding a proposition requires grasping both the conditions under which it can be asserted and the conditions under which it should be denied. We operationalize this principle in the factuality evaluation of atomic formulas using an LLM judge.

First, we generate a natural language verbalization of an atomic formula, then prompt the LLM to generate two statements on the verifiability and refutability of the assertion. The statements are then mapped into the set of truth values $\mathcal{V}_3$ used in weak Kleene logic (Kleene, 1952; Szmuc, 2019), i.e., $\mathfrak{t}$ (true), $\mathfrak{e}$ (undefined), and $\mathfrak{f}$ (false). Weak Kleene truth values allow us to formalize the semantics of LLM-judge output. For example, the SimpleQA grader used in the SimpleQA benchmark (Wei et al., 2024) grades an LLM's answer to a question as either "CORRECT", "INCORRECT", or "NOT ATTEMPTED"; the PreciseWikiQA Question Answerability Prompt in the HalluLens benchmark (Bang et al., 2025) uses "UNVERIFIABLE" instead of "NOT ATTEMPTED". We equate "CORRECT" with $\mathfrak{t}$ and "INCORRECT" with $\mathfrak{f}$; equating $\mathfrak{e}$ with "NOT ATTEMPTED" or "UNVERIFIABLE" is consistent with Kleene's original statement that it indicates "an absence of information" that a given formula is either $\mathfrak{t}$ or $\mathfrak{f}$ (Kleene, 1952, p. 333).

Finally, we pair the weak Kleene truth value $u$ for verifiability with the weak Kleene truth value $v$ for refutability to yield a generalized truth value $\langle u, v \rangle \in \mathcal{V}_3 \times \mathcal{V}_3$. Generalized truth values offer significant advantages over truth values provided by current approaches to factuality evaluation using LLM judges: they enable systematic distinctions between different types of evidence (Shramko and Wansing, 2011; Ferguson, 2021b), provide principled methods for handling inconsistent and incomplete information (Shramko and Wansing, 2011; Szmuc, 2019; Ferguson, 2021b; Correia, 2010), and enhance the explanatory transparency of logical valuations through their structured nature (Shramko and Wansing, 2011; Ferguson, 2021b; Fine, 2016). Figure 3 shows an example of this process in action; a longer example is provided in Appendix C.

**Preliminaries** Let $\Sigma$ be a countable set of tokens and $\Sigma^*$ be the set of finite sequences of tokens $\ulcorner t_0 \ldots t_k \urcorner$, where $t_{0 \le i \le k} \in \Sigma$, $k \in \mathbb{N}$. Given sequences $\sigma, \sigma' \in \Sigma^*$, $\sigma \prec \sigma'$ iff $\sigma$ is a *proper contiguous subsequence* of $\sigma'$. Let $L_{\mathfrak{C}}$ be an LLM trained on a corpus $\mathfrak{C} \in \mathcal{P}(\Sigma^*)$.

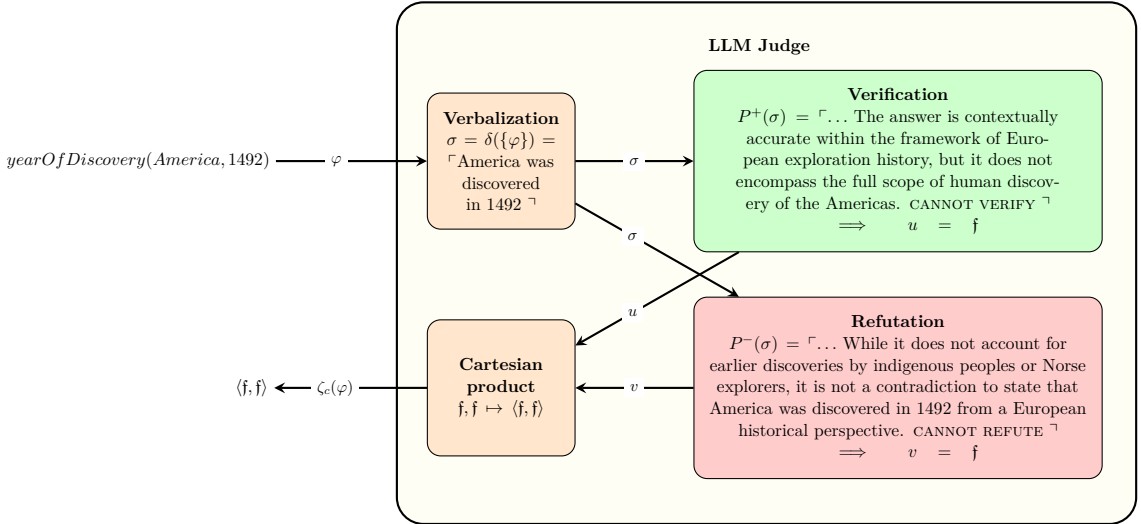

Figure 3: An example of bilateral factuality evaluation $\zeta_c$ as performed by the LLM judge shown in Figure 1. Let $\varphi \in \mathcal{L}_{AT}$ be an assertion that the discovery of America occurred in the year 1492. $\sigma = \delta(\{\varphi\})$ is the verbalization of $\varphi$ (Definition 1). The bilateral factuality evaluation of $\varphi$ by an LLM judge (Definitions 2, 3, and 4) generates the truth value $\zeta_c(\varphi) = \langle \mathfrak{f}, \mathfrak{f} \rangle$. The LLM has in effect identified *incompleteness* in its knowledge — based on differing perspectives on who discovered America, it can neither verify nor refute $\varphi$.

**Definition 1**  *A verbalization function $\delta : \mathcal{P}(\mathcal{L}) \to \Sigma^*$ is a total function that maps a set of formulas to a sequence of tokens.*

In practice, $\delta$ can be implemented through a template-based approach (Ell and Harth, 2014), by prompting an LLM to generate a verbalization (Perevalov and Both, 2024), or by providing formulas directly in the object language syntax.

**Definition 2**  *A verification function $P^+ : \Sigma^* \to \Sigma^*$ prompts $L_{\mathfrak{C}}$ to take a verbalization of an atomic formula $\varphi \in \mathcal{L}_{AT}$ and generate a token sequence $\sigma^+$ that states that $\varphi$ is verified or that it cannot be verified.*

**Definition 3**  *A refutation function $P^- : \Sigma^* \to \Sigma^*$ prompts $L_{\mathfrak{C}}$ to take a verbalization of an atomic formula $\varphi \in \mathcal{L}_{AT}$ and generate a token sequence $\sigma^-$ that states that $\varphi$ is refuted or that it cannot be refuted.*

**Definition 4**  *A bilateral factuality evaluation function $\zeta : \mathcal{L}_{AT} \to \mathcal{V}_3 \times \mathcal{V}_3$ is a total function that given an atomic formula $\varphi \in \mathcal{L}_{AT}$ yields a pair $\langle u, v \rangle$ where:*

$$u = \begin{cases} \mathfrak{t} & \textit{if } \ulcorner \mathsf{VERIFIED} \urcorner \prec P^+(\delta(\varphi)) \\ \mathfrak{f} & \textit{if } \ulcorner \mathsf{CANNOT\ VERIFY} \urcorner \prec P^+(\delta(\varphi)) \\ \mathfrak{e} & \textit{otherwise} \end{cases}$$

$$v = \begin{cases} \mathfrak{t} & \textit{if } \ulcorner \mathsf{REFUTED} \urcorner \prec P^-(\delta(\varphi)) \\ \mathfrak{f} & \textit{if } \ulcorner \mathsf{CANNOT\ REFUTE} \urcorner \prec P^-(\delta(\varphi)) \\ \mathfrak{e} & \textit{otherwise} \end{cases}$$

The "otherwise" cases in Definition 4 reflect LLMs failing to output tokens indicating verification or refutation state, e.g., by failing to follow instructions or timing out during API calls. We use repeated sampling with majority vote (Brown et al., 2024) to determine truth value components; other hallucination mitigation approaches, such as chain-of-verification (Dhuliawala et al., 2023), are also admissible.

The definition of $\zeta$ leaves open the possibility that multiple calls over time might return different truth values. The definition of $\zeta$ allows multiple calls to return different truth values. Analytic tableau reasoning assumes atomic formulas have stable truth values within the reasoning process scope. We ensure this by using a caching version of the bilateral evaluation function $\zeta_c$ where valuations for atomic formulas are persistently and immutably stored. This type of caching is consistent with the range of optimization techniques used in description logics tableau reasoners (Goré and Nguyen, 2007; Nguyen, 2009).

**Definition 5** *A caching bilateral factuality evaluation function $\zeta_c : \mathcal{L}_{AT} \to \mathcal{V}_3 \times \mathcal{V}_3$ is a total function defined as:*

$$\zeta_c(\varphi) = \begin{cases} c(\varphi) & \textit{if } \varphi \in dom(c) \\ \zeta(\varphi) & \textit{otherwise, and } c := c \cup \{(\varphi, \zeta(\varphi))\} \end{cases}$$

*where $c$ is a persistent and immutable cache mapping atomic formulas to truth value pairs.*

Having established a bilateral evaluation, we now show how this enables LLM judges to implement interpretation functions directly.

## 4. LLM-grounded interpretations

We formalize how the bilateral evaluation function $\zeta_c$ is integrated into the definition of an interpretation for **AC**, show that for every LLM-grounded **AC** interpretation there is an equivalent standard **AC** interpretation, and then show that soundness and completeness of the tableau-style analytic calculus **ACrQ** defined in Definition 18 of Ferguson (2021b) is preserved when we adopt an LLM-grounded interpretation.

**Definition 6** *An LLM-grounded **AC** interpretation $\mathcal{I} = \langle \mathbf{C}^{\mathcal{I}}, \mathbf{R}^{\mathcal{I}} \rangle$ is an **AC** interpretation such that for every function $R^{\mathcal{I}} \in \mathbf{R}^{\mathcal{I}}$ and $c_1^{\mathcal{I}}, \ldots, c_n^{\mathcal{I}} \in \mathbf{C}^{\mathcal{I}}$:*

$$R^{\mathcal{I}}(c_1^{\mathcal{I}}, \ldots, c_n^{\mathcal{I}}) = \zeta_c(R(c_1, \ldots, c_n))$$

**Lemma 7 (Stability of LLM-grounded interpretations)** *For any LLM-grounded **AC** interpretation $\mathcal{I}$ and atomic formula $\varphi \in \mathcal{L}_{AT}$:*

1. *$\mathcal{I}(\varphi)$ is well-defined and yields exactly one pair $\langle u, v \rangle \in \mathcal{V}_3 \times \mathcal{V}_3$*
2. *Once computed, $\mathcal{I}(\varphi)$ remains constant throughout the reasoning process*

**Proof** The stability follows directly from Definition 5 of $\zeta_c$. Let $t_0$ be the time at which $\zeta$ is first called to set $c(\varphi)$. If $\zeta_c(\varphi) = \langle u, v \rangle$ at time $t_0$, then for all subsequent calls at $t > t_0$, $\zeta_c(\varphi) = \langle u, v \rangle$. This ensures that once an atomic formula $\varphi$ has been evaluated and the returned pair $\langle u, v \rangle \in \mathcal{V}_3 \times \mathcal{V}_3$ has been cached, all subsequent evaluations will return the same pair from the cache, guaranteeing stability. ∎

Lemma 7 states that logical validity of derivations depends on logical operator structure rather than atomic proposition content, a fundamental principle in formal logic. The tableau method only depends on the truth-functional behavior of the logical connectives, which remains unchanged between standard and LLM-grounded interpretations.

**Lemma 8 (LLM-grounded to standard interpretation mapping)** *For any LLM-grounded* **AC** *interpretation* $\mathcal{I} = \langle \mathbf{C}^{\mathcal{I}}, \mathbf{R}^{\mathcal{I}} \rangle$*, there exists a standard* **AC** *interpretation* $\mathcal{I}'$ *that preserves the semantic behavior of* $\mathcal{I}$ *on all formulas.*

**Proof** Given an LLM-grounded **AC** interpretation $\mathcal{I} = \langle \mathbf{C}^{\mathcal{I}}, \mathbf{R}^{\mathcal{I}} \rangle$, we define a standard **AC** interpretation $\mathcal{I}' = \langle \mathbf{C}^{\mathcal{I}'}, \mathbf{R}^{\mathcal{I}'} \rangle$ such that:

1. $\mathbf{C}^{\mathcal{I}'} = \mathbf{C}^{\mathcal{I}}$
2. $\mathbf{R}^{\mathcal{I}'} = \mathbf{R}^{\mathcal{I}}$
3. For all $R \in \mathbf{R}$, $R^{\mathcal{I}'}(c_1^{\mathcal{I}'}, \ldots, c_n^{\mathcal{I}'}) = R^{\mathcal{I}}(c_1^{\mathcal{I}}, \ldots, c_n^{\mathcal{I}})$
4. For all $c \in \mathcal{C}'$, $c^{\mathcal{I}'} = c^{\mathcal{I}}$

We show that for any formula $\varphi \in \mathcal{L}$, $\mathcal{I}(\varphi) = \mathcal{I}'(\varphi)$ by induction on the complexity of $\varphi$:

- For $\varphi = R(c_1, \ldots c_n) \in \mathcal{L}_{AT}$, then by Definition 6, $\mathcal{I}(\varphi) = \mathcal{I}(R(c_1, \ldots c_n)) = R^{\mathcal{I}}(c_1^{\mathcal{I}}, \ldots, c_n^{\mathcal{I}}) = R^{\mathcal{I}'}(c_1^{\mathcal{I}'}, \ldots, c_n^{\mathcal{I}'}) = \mathcal{I}'(R(c_1, \ldots c_n)) = \mathcal{I}'(\varphi)$.
- For $\varphi = \neg\psi$, by Definition 16, $\mathcal{I}(\neg\psi) = \langle \mathcal{I}_1(\psi), \mathcal{I}_0(\psi) \rangle$ and $\mathcal{I}'(\neg\psi) = \langle \mathcal{I}_1(\psi), \mathcal{I}_0(\psi) \rangle$. By the inductive hypothesis, $\mathcal{I}(\psi) = \mathcal{I}'(\psi)$. Therefore $\mathcal{I}(\neg\psi) = \mathcal{I}'(\neg\psi)$.
- For $\varphi = \psi \wedge \chi$, by Definition 16, $\mathcal{I}(\psi \wedge \chi) = \langle \mathcal{I}_0(\psi) \mathbin{\dot\wedge} \mathcal{I}_0(\chi), \mathcal{I}_1(\psi) \mathbin{\dot\vee} \mathcal{I}_1(\chi) \rangle$ and similarly for $\mathcal{I}'$. By the inductive hypothesis, $\mathcal{I}(\psi) = \mathcal{I}'(\psi)$ and $\mathcal{I}(\chi) = \mathcal{I}'(\chi)$. Therefore $\mathcal{I}(\psi \wedge \chi) = \mathcal{I}'(\psi \wedge \chi)$.

The same arguments apply in the cases of disjunction, restricted universal quantification, and restricted existential quantification, again following Definition 16 in Appendix A. ∎

**Lemma 9 (Standard to LLM-grounded interpretation mapping)** *For any standard* **AC** *interpretation* $\mathcal{I} = \langle \mathbf{C}^{\mathcal{I}}, \mathbf{R}^{\mathcal{I}} \rangle$*, there exists an LLM-grounded* **AC** *interpretation* $\mathcal{I}'$ *that preserves the semantic behavior of* $\mathcal{I}$ *on all formulas.*

**Proof** Let $\mathcal{I} = \langle \mathbf{C}^{\mathcal{I}}, \mathbf{R}^{\mathcal{I}} \rangle$ be a standard **AC** interpretation. Define a key-value store $c_{\mathcal{I}}$ such that for all atomic $R(c_1, \ldots, c_n) \in AT$, $c(R(c_1, \ldots, c_n)) = R^{\mathcal{I}}(c_1^{\mathcal{I}}, \ldots, c_n^{\mathcal{I}})$. Then $c_{\mathcal{I}}$ induces a caching bilateral factuality evaluation function $\zeta_{c_{\mathcal{I}}}$ such that the induced LLM-grounded **AC** interpretation $\mathcal{I}'$ agrees with $\mathcal{I}$ on all atoms. Consequently, an induction along the lines of that in Lemma 8 ensures agreement. ∎

Note that as a corollary of Lemmas 8 and 9, $\Gamma \vDash_{\mathbf{AC}} \varphi$ holds if and only if the inference from $\Gamma$ to $\varphi$ is valid over all LLM-grounded **AC** interpretations. This allows the following two theorems:

**Theorem 10 (Preservation of soundness)** *Let $\Gamma$ be a finite set of formulas and $\varphi$ a formula in* **AC***. If $\Gamma \vdash_{\mathbf{ACrQ}} \varphi$, then $\Gamma \models_{\mathcal{I}} \varphi$ is valid for all LLM-grounded* **AC** *interpretations $\mathcal{I}$.*

**Proof** By Theorem 3 of Ferguson (2021b), if $\Gamma \vdash_{\mathbf{ACrQ}} \varphi$, then $\Gamma \models_{\mathbf{AC}} \varphi$. Therefore $\Gamma \models_{\mathcal{I}} \varphi$, as is the case for any **AC** interpretation, standard or LLM-grounded. ∎

**Theorem 11 (Preservation of completeness)** *Let $\Gamma$ be a finite set of formulas and $\varphi$ a formula in* **AC***. Then if for all LLM-grounded* **AC** *interpretations $\mathcal{I}$, $\Gamma \models_{\mathcal{I}} \varphi$, then $\Gamma \vdash_{\mathbf{ACrQ}} \varphi$.*

**Proof** By Lemma 8, given $\mathcal{I}$, we can construct a standard **AC** interpretation $\mathcal{I}'$ such that if $\Gamma \models_{\mathcal{I}} \varphi$ then $\Gamma \models_{\mathcal{I}'} \varphi$. By the established completeness theorem for standard **AC** interpretations (Theorem 4 of Ferguson (2021b)), if $\Gamma \models_{\mathcal{I}'} \varphi$, then $\Gamma \vdash_{\mathbf{ACrQ}} \varphi$. ∎

## 5. Evaluation

**Data and metrics** To validate the feasibility of LLM-grounded interpretations, we evaluate the bilateral evaluation function $\zeta$ that underlies them, using question/answer pairs from two short-form factuality benchmarks. Given our theoretical results above, this focus on the practicality of atomic formula valuation within our framework provides a proof-of-concept that a Belnap computer of the type described above is feasible.[1] Question/answer pairs in short-form factuality benchmarks are typically factoids providing a question together with a short answer (Figure 3). We use these as an approximation to the verbalization $\delta(\varphi)$ of an atomic formula $\varphi$. We used the short-form factuality benchmarks GPQA (Rein et al., 2023) and SimpleQA (Wei et al., 2024) to create two balanced test datasets (each N=400) for our experiments. Test data preparation and experimental setup are discussed in Appendix D. We evaluated the performance of $\zeta$ over the two datasets using two metrics: *macro F1* against question/answer pairs that the judge did not abstain from, i.e., where the judge provided a valuation of $\zeta(\varphi) = \langle \mathfrak{t}, \mathfrak{f} \rangle$ (i.e., verified and not refuted) or $\zeta(\varphi) = \langle \mathfrak{f}, \mathfrak{t} \rangle$ (i.e., not verified and refuted), and *coverage*, which is the percentage of the total set of question/answer pairs where the judge did not abstain. We also measured the *time taken per evaluation*, and the *number of tokens used per evaluation*.

**LLM judges** We used three flagship LLMs (Llama 4 Maverick, GPT-4o, and Claude 3.5 Sonnet), and three distilled LLMs (Llama 4 Scout, GPT-4o Mini, and Claude 3.5 Haiku). Each LLM was evaluated using three different pairs of prompts: direct prompts that asked for a verification or refutation for the question/answer pair, zero-shot chain-of-thought prompts, and few-shot chain-of-thought prompts. As a baseline, we also used the six models and three prompt types to perform *unilateral* factuality evaluation, which prompts an LLM to simply determine whether a question/answer pair is $\mathfrak{t}$ or $\mathfrak{f}$. The prompt templates used are provided in Appendix B. Standard errors (in parentheses) presented in tables in this section and in Appendix D were estimated by bootstrap resampling: 1000 subsamples of size N=100 were drawn from the classification results within each model type category (Politis and Romano, 1994).

---

1. Code and data for the experiment is available at https://github.com/bradleypallen/bilateral-factuality-evaluation.

| Dataset | Model Type | Bilateral ($\zeta$) | | Unilateral | |
|---|---|---|---|---|---|
| | | Macro F1 | Coverage | Macro F1 | Coverage |
| GPQA | flagship | 0.699 (0.010) | 0.589 (0.008) | 0.633 (0.007) | 1.000 (0.000) |
| | distilled | 0.608 (0.011) | 0.504 (0.008) | 0.559 (0.008) | 1.000 (0.000) |
| SimpleQA | flagship | 0.736 (0.009) | 0.584 (0.008) | 0.657 (0.008) | 1.000 (0.000) |
| | distilled | 0.624 (0.011) | 0.499 (0.008) | 0.570 (0.008) | 1.000 (0.000) |

Table 1: Summary macro F1 (given abstention) and coverage metrics for the bilateral factuality evaluation function $\zeta$ and a baseline unilateral factuality evaluation function.

**Results**   Table 1 compares macro F1 and coverage between the unilateral and bilateral evaluations across the two datasets, grouped by whether a model's type was flagship or distilled, and Table 2 does the same for mean time of execution and mean tokens used. Table 3 summarizes the distribution of truth values produced in bilateral evaluation. Detailed breakouts are shown in the tables in Appendix D. Our key findings are as follows.

1. ***Bilateral evaluation macro F1 outperforms unilateral evaluation*** ($p < 0.01$) ***at the cost of lower coverage***. The mean difference between bilateral and unilateral macro F1 is 0.062 and for coverage is -0.456.

2. ***Flagship models outperform distilled models*** ($p < 0.01$). Table 1 shows that this is the case for both unilateral and bilateral approaches, though the difference is more pronounced with bilateral evaluation (0.091 on the GPQA dataset, 0.112 on the SimpleQA dataset) versus unilateral evaluation (0.074 on the GPQA dataset, 0.087 on the SimpleQA dataset).

3. ***Bilateral evaluation is more expensive than unilateral evaluation*** ($p < 0.01$). Table 2 shows that bilateral evaluation takes roughly twice as much time and twice as many tokens as unilateral evaluation. However, evaluation times vary widely, with GPT-4o Mini using direct prompting taking a mean of 2.5 seconds, and Llama 4 Scout using zero-shot prompting taking a mean of 43.4 seconds. Token consumption scales predictably, from up to a mean of 2,008.6 tokens with direct prompting, and up to a mean of 6,704.7 tokens with few-shot prompting.

4. ***Inconsistency occurs significantly more frequently than incompleteness*** ($p < 0.05$). Table 3 shows that bilateral judge models more frequently abstain by assigning $\langle \mathfrak{t}, \mathfrak{t} \rangle$ (both verified and refuted) as opposed to $\langle \mathfrak{f}, \mathfrak{f} \rangle$ (neither verified nor refuted).

**Limitations**   We have provided a theoretical framework for integrating an LLM with a paraconsistent reasoner, and demonstrated the feasibility of providing LLM-grounded valuations of atomic formulas, but there is as yet ***no complete implementation of the Belnap computer***. One approach, described by Maier et al. (2013), involves mapping a knowledge base expressed in a paraconsistent version of $\mathcal{SROIQ}$ into a knowledge base expressed in classical $\mathcal{SROIQ}$, which then enables reasoning using an existing description logic reasoner such as Pellet (Sirin et al., 2007).

The computational complexity depends on the number of atomic formulas requiring evaluation, which could be exponential in the worst case. Each atomic formula requires multiple API calls ($2k$ calls for $k$-sample majority voting), making ***inference API costs***

| Dataset | Model Type | Bilateral ($\zeta$) | | Unilateral | |
| --- | --- | --- | --- | --- | --- |
| | | Time (s) | Tokens | Time (s) | Tokens |
| GPQA | flagship | 36.747 (0.281) | 4781.663 (45.878) | 12.411 (0.140) | 2212.766 (27.182) |
| | distilled | 34.771 (0.224) | 4731.672 (41.661) | 13.439 (0.095) | 2532.153 (26.615) |
| SimpleQA | flagship | 32.789 (0.274) | 4163.807 (37.704) | 12.256 (0.140) | 2100.465 (25.634) |
| | distilled | 30.310 (0.253) | 3964.037 (39.991) | 11.424 (0.120) | 2167.419 (28.044) |

Table 2: Summary execution time (in seconds) and total tokens used for the bilateral factuality evaluation function $\zeta$ and a baseline unilateral factuality evaluation function.

| Dataset | Model Type | $\langle t, t \rangle$ | $\langle t, f \rangle$ | $\langle f, t \rangle$ | $\langle f, f \rangle$ |
| --- | --- | --- | --- | --- | --- |
| GPQA | flagship | 0.301 (0.008) | 0.211 (0.007) | 0.378 (0.008) | 0.110 (0.005) |
| | distilled | 0.299 (0.007) | 0.192 (0.006) | 0.312 (0.007) | 0.197 (0.006) |
| SimpleQA | flagship | 0.310 (0.007) | 0.228 (0.007) | 0.357 (0.008) | 0.106 (0.005) |
| | distilled | 0.301 (0.007) | 0.233 (0.007) | 0.266 (0.007) | 0.200 (0.006) |

Table 3: Summary truth value distributions for the bilateral factuality evaluation function $\zeta$. Of note is the fact that the models evaluated did not produce any truth values where $u = \mathfrak{e}$ or $v = \mathfrak{e}$ during the evaluation.

***and latency immediate practical bottlenecks***. Yet, we expect that caching in $\zeta_c$ will amortize costs across repeated evaluations, and that standard tableau optimization techniques will help keep the overall complexity manageable. Empirical validation of these complexity expectations remains future work.

## 6. Conclusion and future work

We described a novel approach to logical reasoning using LLMs with several key contributions. We defined a bilateral approach to factuality evaluation that identifies gaps and contradictions in LLM parametric knowledge. We introduced the concept of LLM-grounded interpretations that integrate an LLM directly into the formal semantics of the underlying logic while preserving its soundness and completeness. We provided empirical evidence that this formal framework can be realized in practice, providing a path to enabling LLMs to serve as broad-coverage knowledge sources for logical reasoners.

In future work, we plan to implement a translation (Arieli and Denecker, 2003) between **AC** and a classical description logic such as $\mathcal{SROIQ}$, allowing us to evaluate a fully-integrated reasoner using an LLM-grounded interpretation. We also plan to extend the above approach based on work in the area of generalized truth values (Shramko and Wansing, 2005; Hornischer, 2025) to provide a multi-valued semantics for modeling factuality in LLMs, with the goal of improving the theoretical framework and metrics for factuality evaluation.

## Acknowledgements

This work was partially supported by the EU's Horizon Europe research and innovation programme within the ENEXA project (grant Agreement no. 101070305). Particular thanks

are due to Fabian Hoppe, Levin Hornischer, Jan-Christoph Kalo, and Lise Stork for discussions and perceptive observations that enriched our research.

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

## Appendix A. Angell's logic of analytic containment AC

Below we summarize the definitions of the object language, truth functions, and interpretations for the version of **AC** presented in greater detail in Ferguson (2021b).

### A.1. Object language

**Definition 12** *Let $\mathcal{L}$ be a first-order language built from a countable set $\mathcal{C}$ of constants, a countable set of variables $\mathcal{V}$, a countable set $\mathcal{R}$ of relation symbols, the Boolean connectives $\neg$, $\wedge$, and $\vee$, restricted universal and existential quantifiers $\forall$ and $\exists$, and round parentheses (as used in complex formulas) and square brackets (as used in quantified formulas) as auxiliary symbols. If $R \in \mathcal{R}$ and $c_1, \ldots, c_n \in \mathcal{C}$, then $R(c_1, \ldots, c_n)$ is an atomic formula. Let $\mathcal{L}_{AT}$ be the set of atomic formulas. The formulas of $\mathcal{L}$ are the elements of $\mathcal{L}_{AT}$, together with the following, where $\varphi, \psi \in \mathcal{L}$ and $x \in \mathcal{V}$:*

$$\neg\varphi \mid (\varphi \wedge \psi) \mid (\varphi \vee \psi) \mid [\forall x \varphi(x)]\psi(x) \mid [\exists x \varphi(x)]\psi(x)$$

### A.2. Truth functions

**Definition 13** *The* weak Kleene truth tables *over the set of truth values $\mathcal{V}_3 = \{\mathfrak{t}, \mathfrak{e}, \mathfrak{f}\}$ are:*

| $\neg$ | |
|---|---|
| $\mathfrak{t}$ | $\mathfrak{f}$ |
| $\mathfrak{e}$ | $\mathfrak{e}$ |
| $\mathfrak{f}$ | $\mathfrak{t}$ |

| $\wedge$ | $\mathfrak{t}$ | $\mathfrak{e}$ | $\mathfrak{f}$ |
|---|---|---|---|
| $\mathfrak{t}$ | $\mathfrak{t}$ | $\mathfrak{e}$ | $\mathfrak{f}$ |
| $\mathfrak{e}$ | $\mathfrak{e}$ | $\mathfrak{e}$ | $\mathfrak{e}$ |
| $\mathfrak{f}$ | $\mathfrak{f}$ | $\mathfrak{e}$ | $\mathfrak{f}$ |

| $\vee$ | $\mathfrak{t}$ | $\mathfrak{e}$ | $\mathfrak{f}$ |
|---|---|---|---|
| $\mathfrak{t}$ | $\mathfrak{t}$ | $\mathfrak{e}$ | $\mathfrak{t}$ |
| $\mathfrak{e}$ | $\mathfrak{e}$ | $\mathfrak{e}$ | $\mathfrak{e}$ |
| $\mathfrak{f}$ | $\mathfrak{t}$ | $\mathfrak{e}$ | $\mathfrak{f}$ |

The weak Kleene truth tables for conjunction and disjunction induce the truth functions $\dot{\wedge}$ and $\dot{\vee}$, respectively.

**Definition 14** *The* restricted Kleene quantifier functions $\dot{\forall}$ and $\dot{\exists}$ are mappings from sets of truth values to truth values such that:*

$$\dot{\exists}(X) = \begin{cases} \mathfrak{t} & \text{if } \langle \mathfrak{t}, \mathfrak{t} \rangle \in X \\ \mathfrak{e} & \text{if for all } \langle u, v \rangle, \text{ either } u = \mathfrak{e} \text{ or } v = \mathfrak{e} \\ \mathfrak{f} & \text{if } \langle \mathfrak{t}, \mathfrak{t} \rangle \notin X \text{ and for some } \langle u, v \rangle \in X, u \neq \mathfrak{e} \text{ and } v \neq \mathfrak{e} \end{cases}$$

$$\dot{\forall}(X) = \begin{cases} \mathfrak{t} & \text{if } \langle \mathfrak{t}, \mathfrak{f} \rangle, \langle \mathfrak{t}, \mathfrak{e} \rangle \notin X \text{ and for some } \langle u, v \rangle \in X, u \neq \mathfrak{e} \text{ and } v \neq \mathfrak{e} \\ \mathfrak{e} & \text{if for all } \langle u, v \rangle \in X, \text{ either } u = \mathfrak{e} \text{ or } v = \mathfrak{e} \\ \mathfrak{f} & \text{if } \{\langle \mathfrak{t}, \mathfrak{t} \rangle, \langle \mathfrak{t}, \mathfrak{e} \rangle\} \cap X \neq \emptyset \text{ and for some } \langle u, v \rangle \in X, \text{ either } u = \mathfrak{e} \text{ or } v = \mathfrak{e} \end{cases}$$

### A.3. Interpretations

**Definition 15** *An* **AC** interpretation $\mathcal{I}$ is a pair $\langle \mathbf{C}^{\mathcal{I}}, \mathbf{R}^{\mathcal{I}} \rangle$ where $\mathbf{C}^{\mathcal{I}}$ is a domain of individuals and $\mathbf{R}^{\mathcal{I}}$ is a set of functions where $\mathcal{I}$ assigns:*

- *every constant $c \in \mathbf{C}$ an individual $c^{\mathcal{I}} \in \mathbf{C}^{\mathcal{I}}$*
- *every $n$-ary predicate $R$ a function $R^{\mathcal{I}} : (\mathbf{C}^{\mathcal{I}})^n \to \mathcal{V}_3 \times \mathcal{V}_3$*

**Definition 16** *An* **AC** *interpretation $\mathcal{I}$ induces a map $\mathcal{I} : \mathcal{L} \to \mathcal{V}_3 \times \mathcal{V}_3$ as follows, where $\mathcal{I}_0$ and $\mathcal{I}_1$ project the first and second coordinates respectively:*

- *For atomic formulas $R(c_1, \ldots, c_n) \in \mathcal{L}_{AT}$, $\mathcal{I}(\varphi) = R^{\mathcal{I}}(c_1^{\mathcal{I}}, \ldots, c_n^{\mathcal{I}})$*
- $\mathcal{I}(\neg\varphi) = \langle \mathcal{I}_1(\varphi), \mathcal{I}_0(\varphi) \rangle$
- $\mathcal{I}(\varphi \wedge \psi) = \langle \mathcal{I}_0(\varphi) \dot{\wedge} \mathcal{I}_0(\psi), \mathcal{I}_1(\varphi) \dot{\vee} \mathcal{I}_1(\psi) \rangle$
- $\mathcal{I}(\varphi \vee \psi) = \langle \mathcal{I}_0(\varphi) \dot{\vee} \mathcal{I}_0(\psi), \mathcal{I}_1(\varphi) \dot{\wedge} \mathcal{I}_1(\psi) \rangle$

- $\mathcal{I}([\forall x \varphi(x)]\psi(x)) = \langle \dot{\forall}(\{\mathcal{I}_0(\varphi(c)), \mathcal{I}_0(\psi(c)) \mid c \in \mathcal{C}\}), \dot{\exists}(\{\mathcal{I}_0(\varphi(c)), \mathcal{I}_1(\psi(c)) \mid c \in \mathcal{C}\})\rangle$
- $\mathcal{I}([\exists x \varphi(x)]\psi(x)) = \langle \dot{\exists}(\{\mathcal{I}_0(\varphi(c)), \mathcal{I}_0(\psi(c)) \mid c \in \mathcal{C}\}), \dot{\forall}(\{\mathcal{I}_0(\varphi(c)), \mathcal{I}_1(\psi(c)) \mid c \in \mathcal{C}\})\rangle$

**Definition 17** *Given an* **AC** *interpretation* $\mathcal{I}$, *validity with respect to* $\mathcal{I}$ *is defined as truth preservation, i.e.*

$$\Gamma \models_{\mathcal{I}} \varphi \text{ if for all instances of } \mathcal{I} \text{ such that } \forall \psi \in \Gamma \, \mathcal{I}_0(\psi) = \mathfrak{t}, \, \mathcal{I}_0(\varphi) = \mathfrak{t}.$$

# Appendix B. Prompts

## B.1. Direct verification template

```
Determine whether the following answer to the given question is correct.
Conclude with a single line containing ONLY one of these two phrases:
VERIFIED
CANNOT VERIFY

Question: {question}
Proposed answer: {answer}
```

## B.2. Direct refutation template

```
Determine whether the following answer to the given question can be refuted.
Conclude with a single line containing ONLY one of these two phrases:
REFUTED
CANNOT REFUTE

Question: {question}
Proposed answer: {answer}
```

## B.3. Zero-shot verification template

```
I'll provide you with a question and its proposed answer.
Your task is to verify whether this answer is correct by following these steps:

1. Analyze the exact meaning of both the question and answer,
identifying any key terms that need clarification.
2. Establish specific conditions that would make this answer true for this question.
3. Provide direct evidence supporting the answer, including specific facts, examples,
or authoritative references that confirm its accuracy.
4. Test if the answer remains valid across all contexts where the question applies,
noting any limitations or exceptions.
5. Check for consistency with established knowledge in the relevant domain.

Based on your analysis, determine whether the answer is verified and explain
your reasoning with specific supporting evidence.
Your goal is not to find fault but to determine if positive
evidence exists to confirm the answer.

After your complete analysis, conclude with a single line containing
ONLY one of these two phrases:
VERIFIED
CANNOT VERIFY

Question: {question}
Proposed answer: {answer}
```

## B.4. Zero-shot refutation template

```
I'll provide you with a question and its proposed answer.
Your task is to determine if this answer can be refuted by following these steps:

1. Analyze the exact meaning of both the question and the proposed answer.
2. Identify what specific conditions would need to be true for this answer to be false
(not merely the absence of evidence).
3. Search for direct counterexamples or contradicting evidence that
actively demonstrates why the answer is incorrect.
4. Construct specific scenarios where the answer fails to hold true,
even if the question's premises are accepted.
5. Identify any logical inconsistencies, factual errors, or category mistakes
within the answer.

Focus on building an affirmative case for why the answer is incorrect,
rather than simply noting a lack of supporting evidence.
Provide specific counterevidence and explain precisely
how it contradicts the proposed answer.

After your complete analysis, conclude with a single line containing
ONLY one of these two phrases:
REFUTED
CANNOT REFUTE

Question: {question}
Proposed answer: {answer}
```

## B.5. Few-shot verification template

```
I'll provide you with a question and its proposed answer.
Your task is to verify whether this answer is correct by following these steps:

1. Analyze the exact meaning of both the question and answer, identifying any key terms that need clarification.
2. Establish specific conditions that would make this answer true for this question.
3. Provide direct evidence supporting the answer, including specific facts, examples, or authoritative references that confirm
its accuracy.
4. Test if the answer remains valid across all contexts where the question applies, noting any limitations or exceptions.
5. Check for consistency with established knowledge in the relevant domain.

Based on your analysis, determine whether the answer is verified and explain your reasoning with specific supporting evidence.
Your goal is not to find fault but to determine if positive evidence exists to confirm the answer.

Here are some examples of how to approach verification:

Example 1:
Question: Who was the first man to walk on the Moon?
Proposed answer: Neil Armstrong
Let's verify this answer systematically:
Key Terms Analysis:
"First man to walk on the Moon" requires precise historical verification
Specific date and mission details are crucial
Conditions for Verification:
Must be the first human to physically step onto lunar surface
Must have documented evidence of this historic moment
Must be part of a specific NASA mission
Direct Evidence:
Neil Armstrong stepped onto the Moon's surface on July 20, 1969
During Apollo 11 mission
Famous quote: "That's one small step for man, one giant leap for mankind"
Extensive NASA documentation confirms this
Broadcast live on global television
Verified by multiple independent sources including NASA, scientific records, and historical archives
Context and Limitations:
Specifically refers to first human, not first spacecraft
Specifically lunar surface, not orbital passage
Confirmed during Cold War space race
Part of NASA's Apollo program
Consistency Check:
Matches scientific and historical consensus
No credible alternative claims exist
Extensively documented in multiple international records
This answer meets all verification criteria with robust, multi-source evidence.
VERIFIED

Example 2:
Question: Chocolate is made from material
Proposed answer: Sugar
Let's analyze this systematically:
Key Term Analysis:
"Chocolate" refers to a food product derived from cacao/cocoa beans
"Material" suggests the fundamental ingredient or component
"Sugar" is a sweetener, not the primary material of chocolate
Conditions for Accuracy:
Sugar is an ingredient in chocolate
But sugar is NOT the primary material of chocolate
Chocolate's primary material is cocoa/cacao solids
Direct Evidence:
Chocolate is primarily made from cocoa beans processed into cocoa solids
Cocoa solids come from cacao tree seeds/beans
Sugar is added as a sweetener, not the base material
Chocolate composition typically includes:

Cocoa solids (primary material)
Cocoa butter
Sugar (secondary ingredient)
Milk (in milk chocolate)

Context Testing:
In all chocolate production processes, cocoa is the fundamental material
Sugar is always a supplementary ingredient, not the base material
Domain Consistency:
Culinary and food science consistently define cocoa/cacao as chocolate's primary material
Based on comprehensive analysis, the proposed answer is incorrect.
CANNOT VERIFY

After your complete analysis, conclude with a single line containing ONLY one of these two phrases:
VERIFIED
CANNOT VERIFY

Question: {question}
Proposed answer: {answer}
```

## B.6. Few-shot refutation template

```
I'll provide you with a question and its proposed answer.
Your task is to determine if this answer can be refuted by following these steps:

1. Analyze the exact meaning of both the question and the proposed answer.
2. Identify what specific conditions would need to be true for this answer to be false (not merely the absence of evidence).
3. Search for direct counterexamples or contradicting evidence that actively demonstrates why the answer is incorrect.
4. Construct specific scenarios where the answer fails to hold true, even if the question's premises are accepted.
5. Identify any logical inconsistencies, factual errors, or category mistakes within the answer.

Focus on building an affirmative case for why the answer is incorrect, rather than simply noting a lack of supporting evidence.
Provide specific counterevidence and explain precisely how it contradicts the proposed answer.

Here are some examples of how to approach refutation:

Example 1:
Question: Are penguins birds?
Proposed answer: No
Let's analyze this systematically:
Meaning Analysis:
Question asks about the taxonomic classification of penguins
Proposed answer claims penguins are NOT birds
Conditions for Falsity:
Penguins must meet standard biological criteria for birds
Must share key avian characteristics
Counterevidence:
Penguins have ALL classic bird characteristics:

Feathered body
Lay eggs
Warm-blooded
Have beaks
Descended from dinosaur lineage
Classified in scientific taxonomy under Class Aves
Specifically, penguins belong to the order Sphenisciformes, which is a recognized bird order
Biological and genetic evidence conclusively places penguins within bird classification

Specific Scenarios Contradicting Answer:
Penguins have wing-like flippers adapted for swimming
They have respiratory and skeletal structures identical to other bird species
Genetic sequencing confirms their bird lineage
Logical Inconsistencies:
Rejecting penguins as birds would require rejecting fundamental biological classification systems
No scientific basis exists for excluding penguins from bird category
REFUTED

Example 2:
Question: Who was the first man to walk on the Moon?
Proposed answer: Neil Armstrong
Let's analyze this systematically:
Meaning Analysis:
Question seeks the definitive first human male to set foot on lunar surface
Proposed answer: Neil Armstrong (Apollo 11 mission, July 20, 1969)
Potential Conditions for Falsity:
Documented evidence of another person walking on Moon before Armstrong
Proof that Armstrong was not actually the first
Historical record showing a different individual preceded him
Counterevidence Search:
No credible historical evidence exists contradicting Armstrong's first Moon walk
NASA records and global documentation consistently confirm Armstrong as first
Extensive photographic and video evidence supports this claim
Scenario Testing:
No alternative scenarios emerge that could plausibly replace Armstrong's achievement
Extensive verification by multiple nations and independent researchers confirms his primacy
Logical Consistency Check:
Armstrong's Moon walk is extensively documented
Multiple witnesses and technological records corroborate the event
No logical inconsistencies detected in the claim
The proposed answer is completely accurate and supported by overwhelming historical evidence.
CANNOT REFUTE

After your complete analysis, conclude with a single line containing ONLY one of these two phrases:
REFUTED
CANNOT REFUTE

Question: {question}
Proposed answer: {answer}
```

## B.7. Prompt template for generating negative examples for SimpleQA-derived benchmark

```
You are an expert synthetic data generator. Your task is to generate three plausible but
incorrect answers to a given question that will serve as challenging distractors.

Guidelines for generating high-quality wrong answers:
1. Each answer must be factually incorrect but highly plausible within the context
   - Draw from the same domain/topic as the correct answer
   - Use answers that could reasonably be mistaken for the truth
   - Avoid obviously wrong or nonsensical options

2. Strictly match the answer type and format
   - For dates: Use the same date format and plausible timeframe
   - For people: Match profession, era, and relevance
   - For numbers: Stay within reasonable orders of magnitude
   - For places: Use locations of similar type/scale

3. Ensure clear differentiation
   - Make each wrong answer distinct from the correct answer
   - Avoid overlap between wrong answers
   - Space out numerical answers appropriately

4. Maintain consistent specificity
   - Match the level of detail in the correct answer
   - If the answer is detailed, provide equally detailed wrong options
   - If the answer is brief, keep wrong answers similarly concise

5. Optimize for realism and difficulty
   - Wrong answers should feel natural, not contrived
   - Target common misconceptions when possible
   - Make the options challenging enough to test real knowledge
   - Avoid answers that are too obviously incorrect

Example 1:
Question: Who was the first president of the United States?
Answer: George Washington
Wrong Answers:
- John Adams
- Thomas Jefferson
- Benjamin Franklin
Reason: All are founding fathers but not the first president

Example 2:
Question: What is the largest planet in our solar system?
Answer: Jupiter
Wrong Answers:
- Saturn
- Neptune
- Uranus
Reason: All are gas giant planets, but smaller than Jupiter

Example 3:
Question: Who wrote Romeo and Juliet?
Answer: William Shakespeare
Wrong Answers:
- Christopher Marlowe
- Ben Jonson
- John Webster
Reason: All are prominent Elizabethan playwrights

Return only three wrong answers as a list in JSON format with the following requirements:
- Each wrong answer should be a string
- The output should be a single JSON object with key "negative_answers"
- The value should be an array of exactly 3 wrong answers
- No explanations or additional text should be included
- The answers should maintain consistent formatting with the correct answer

Example format:
{{
    "negative_answers": ["Wrong Answer 1", "Wrong Answer 2", "Wrong Answer 3"]
}}
```

## B.8. Direct prompt template for unilateral factuality evaluation

```
Determine whether the following answer to the given question is TRUE or FALSE.

Conclude with a single line containing ONLY one of these two words:
TRUE
FALSE

Question: {question}
Proposed answer: {answer}
```

## B.9. Zero-shot prompt template for unilateral factuality evaluation

```
I'll provide you with a question and its proposed answer.
Your task is to evaluate whether this answer is correct by following these steps:

1. Analyze the exact meaning of both the question and answer, identifying any key terms that need clarification.
2. Establish specific conditions that would make this answer true for this question.
3. Provide direct evidence supporting the answer, including specific facts, examples, or
authoritative references that confirm its accuracy.
4. Test if the answer remains valid across all contexts where the question applies, noting any limitations or exceptions.
5. Check for consistency with established knowledge in the relevant domain.
6. Search for direct counterexamples or contradicting evidence that actively demonstrates why the answer is incorrect.
7. Construct specific scenarios where the answer fails to hold true, even if the question's premises are accepted.
8. Identify any logical inconsistencies, factual errors, or category mistakes within the answer.

After analyzing the question and answer, provide a single line containing ONLY one of these two words:
TRUE
FALSE

Question: {question}
Proposed answer: {answer}
```

## B.10. Few-shot prompt template for unilateral factuality evaluation

```
I'll provide you with a question and its proposed answer.
Your task is to evaluate whether this answer is correct by following these steps:

1. Analyze the exact meaning of both the question and answer, identifying any key terms that need clarification.
2. Establish specific conditions that would make this answer true for this question.
3. Provide direct evidence supporting the answer, including specific facts, examples, or
authoritative references that confirm its accuracy.
4. Test if the answer remains valid across all contexts where the question applies, noting any limitations or exceptions.
5. Check for consistency with established knowledge in the relevant domain.
6. Search for direct counterexamples or contradicting evidence that actively demonstrates why the answer is incorrect.
7. Construct specific scenarios where the answer fails to hold true, even if the question's premises are accepted.
8. Identify any logical inconsistencies, factual errors, or category mistakes within the answer.

Here are examples of how to approach evaluation:

Example 1:
Question: Who was the first man to walk on the Moon?
Proposed answer: Neil Armstrong
Analyze the question and answer:
Question: "Who was the first man to walk on the Moon?" This is a straightforward factual question seeking the identity of
the first human to set foot on the lunar surface.
Proposed answer: "Neil Armstrong" This is a name, presumably offered as the answer to the question.
Establish conditions for truth:
The answer is true if Neil Armstrong was indeed the first human to walk on the Moon.
Provide supporting evidence:
Historical records, NASA documentation, and countless reliable sources confirm that Neil Armstrong was the first person to
walk on the Moon on July 20, 1969, during the Apollo 11 mission.
Test validity across contexts:
The answer holds true in all historical contexts related to the first Moon landing.
Check for consistency with established knowledge:
The answer aligns perfectly with established historical and scientific knowledge.
Search for counterexamples:
There are no credible counterexamples. No other individual is historically recognized as the first person to walk on the Moon.
Construct failure scenarios:
There are no scenarios where the answer fails, assuming the question refers to the generally accepted historical event.
Identify logical inconsistencies:
There are no logical inconsistencies or factual errors.
TRUE

Example 2:
Question: What is the main ingredient in chocolate?
Proposed answer: Sugar
Analyze the question and answer:
Question: "Chocolate is made from material" - This is an incomplete sentence. The question is implicitly asking "What material is
chocolate made from?" or "What is a key material used to make chocolate?".
Proposed answer: "Sugar" - This suggests that sugar is the material chocolate is made from.
Establish conditions for truth:
The answer would be true if sugar was the only ingredient in chocolate, or
if the question was interpreted as "Is sugar a material used to make chocolate?".
Provide supporting evidence:
Sugar is a common and significant ingredient in most chocolate recipes.
Test validity across contexts:
This answer fails in many contexts. Chocolate is not only made from sugar.
Check for consistency with established knowledge:
Chocolate is made from cacao beans, sugar, and often other ingredients like milk solids, cocoa butter, lecithin, and flavorings.
Search for counterexamples:
Dark chocolate often contains a higher percentage of cacao and less sugar.
Sugar-free chocolate exists, using artificial sweeteners instead.
Cacao beans are essential for chocolate, and chocolate cannot be made without them.
Construct failure scenarios:
Imagine a recipe for 100% cacao chocolate. It would contain no sugar.
Imagine a sugar-free chocolate bar. It would contain no sugar.
Identify logical inconsistencies:
The answer implies sugar is the only ingredient, which is false.
FALSE

Question: {question}
Proposed answer: {answer}
```

## Appendix C. Example

To illustrate bilateral evaluation, we present an example with statements about penguins in the context of a knowledge base with a universally quantified statement that all birds can fly:

1. $\mathbf{C} = \{penguin, eagle, sparrow, ...\}$

2. $\varphi_0 = [\,\forall x\, bird(x)\,]\, flies(x)$
3. $\varphi_1 = \mathrm{bird(penguin)}$
4. $\varphi_2 = \mathrm{flies(penguin)}$
5. $\varphi_3 = \neg\varphi_2$
6. $\delta(\varphi_1) = \lceil$Penguins are birds$\rceil$
7. $P^+(\delta(\varphi_1)) = \lceil\ldots$Penguins are scientifically classified as birds. They belong to the family Spheniscidae$\ldots$Conclusion: VERIFIED$\rceil$
8. $P^-(\delta(\varphi_1)) = \lceil\ldots$All evolutionary biologists classify penguins as birds. This is supported by molecular evidence, fossil records, and anatomical features. There is no reasonable alternative classification. Conclusion: CANNOT REFUTE$\rceil$
9. $\zeta(\varphi_1) = \langle t, f\rangle$
10. $\delta(\varphi_2) = \lceil$Penguins fly$\rceil$
11. $P^+(\delta(\varphi_2)) = \lceil\ldots$While penguins have wings, they cannot achieve aerial flight. Their wings are adapted for swimming rather than flying. They flap their wings underwater to "fly" through water. From a strict biological perspective, penguins do not fly through air. Conclusion: CANNOT VERIFY$\rceil$
12. $P^-(\delta(\varphi_2)) = \lceil\ldots$Penguins are flightless birds. Their wings have evolved into flippers for aquatic propulsion rather than aerial flight. This is well-established in ornithology. Conclusion: REFUTED$\rceil$
13. $\zeta(\varphi_2) = \langle f, t\rangle$
14. $\mathcal{I}(\varphi_0) = \langle\dot\forall(\{\mathcal{I}_0(\mathrm{bird}(c)), \mathcal{I}_0(\mathrm{flies}(c)) \mid c \in \mathbf{C}\}), \dot\exists(\{\mathcal{I}_0(\mathrm{bird}(c)), \mathcal{I}_1(\mathrm{flies}(c)) \mid c \in \mathbf{C}\})\rangle$
    $= \langle f, t\rangle$
15. $\mathcal{I}(\varphi_3) = \langle\mathcal{I}_1(\varphi_2), \mathcal{I}_0(\varphi_2)\rangle = \langle t, f\rangle$

This bilateral evaluation reveals the inconsistency. The universal statement $\varphi_0$ evaluates to false when considering penguins, and $\varphi_3$ evaluates to true, but both statements can coexist in **AC** without causing explosion. This demonstrates how the system handles the classic penguin problem through paraconsistent reasoning.

## Appendix D. Experiments

### D.1. Datasets

We used the short-form factuality benchmarks GPQA (Rein et al., 2023) and SimpleQA (Wei et al., 2024) to create the benchmarks for our experiments. GPQA consists of 448 multiple-choice questions, written by domain experts in biology, physics, and chemistry. SimpleQA consists of 4,326 question/answer pairs addressing a range of general topic areas, including history, science and technology, art, geography, TV shows, and video games. From these two benchmarks we created a balanced set of positive and negative examples. From SimpleQA, we sampled without replacement 200 question/answer pairs to be positive examples, and 200 questions to be negative examples, where we substituted false answers synthetically generated using GPT-4o Mini using the prompt shown in Appendix B.7. From GPQA, we sampled 200 existing question/answer pairs as positive examples, and 200 questions paired with the first incorrect answer for that question provided as part of the dataset.

### D.2. Experimental setup

Following Wei et al. (2024), we evaluated our implementation of $\zeta$ on a selective classification with a binary abstention task (El-Yaniv and Wiener, 2010) using the above two datasets,

measuring an LLM judge's grading of a given question/answer pair. The standard pattern in LLM judges in factuality evaluation is to prompt the LLM judge to evaluate the answer to the question as either correct, incorrect, or not attempted. This, again, has a natural mapping to the values of $\mathcal{V}_3$; to derive a single truth value $v \in \mathcal{V}_3$ for the evaluation, we use the following projection $p : \mathcal{V}_3 \times \mathcal{V}_3 \to \mathcal{V}_3$ such that for a pair $\langle u, v \rangle$:

$$
p(\langle u, v \rangle) = \begin{cases} \mathfrak{t} & \text{if } \langle u, v \rangle = \langle \mathfrak{t}, \mathfrak{f} \rangle \\ \mathfrak{f} & \text{if } \langle u, v \rangle = \langle \mathfrak{f}, \mathfrak{t} \rangle \\ \mathfrak{e} & \text{otherwise} \end{cases}
$$

Calls to the public inference APIs for the models used a temperature of 0.1. Repeated sampling (N=3) with majority vote was used in both the verification and refutation processes. Statistical significance was assessed using paired t-tests (`ttest_rel` from the `scipy.stats` Python package) to compare performance metrics between different model and prompt combinations. The experiments were conducted in the first half of May 2025 using calls to the public inference APIs for each of the models.

## D.3. Performance metrics

| Judge Model | Prompt | Macro F1 | Coverage | Time (s) | Tokens |
|---|---|---|---|---|---|
| Claude 3.5 Sonnet | direct | 0.712 (0.023) | 0.748 (0.02) | 34.22 (6.08) | 2914.80 (778.99) |
| | zero | 0.738 (0.029) | 0.542 (0.024) | 53.91 (6.41) | 4863.48 (731.27) |
| | few | 0.716 (0.028) | 0.54 (0.023) | 52.92 (6.41) | 8079.40 (745.71) |
| Claude 3.5 Haiku | direct | 0.578 (0.027) | 0.778 (0.02) | 30.52 (4.09) | 2641.07 (704.48) |
| | zero | 0.648 (0.034) | 0.412 (0.023) | 43.12 (3.60) | 4221.73 (685.85) |
| | few | 0.604 (0.034) | 0.438 (0.024) | 47.44 (4.69) | 7673.44 (700.47) |
| Llama 4 Maverick | direct | **0.774** (0.021) | **0.852** (0.016) | 65.91 (52.53) | 6225.19 (1526.74) |
| | zero | 0.765 (0.025) | 0.618 (0.022) | 75.95 (44.90) | 7492.54 (1357.27) |
| | few | 0.751 (0.023) | 0.805 (0.018) | 65.08 (41.43) | 9945.70 (1444.02) |
| Llama 4 Scout | direct | 0.702 (0.025) | 0.712 (0.021) | 63.24 (28.92) | 5403.00 (1833.85) |
| | zero | 0.712 (0.031) | 0.46 (0.024) | 93.43 (51.27) | 7062.05 (1430.96) |
| | few | 0.694 (0.027) | 0.642 (0.022) | 69.27 (37.27) | 9540.89 (1362.86) |
| GPT-4o | direct | 0.592 (0.027) | 0.705 (0.021) | **5.61** (8.41) | **1133.92** (556.51) |
| | zero | 0.603 (0.035) | 0.48 (0.022) | 36.29 (9.44) | 6041.24 (1256.13) |
| | few | 0.69 (0.031) | 0.518 (0.023) | 42.01 (18.48) | 8662.78 (1398.93) |
| GPT-4o Mini | direct | 0.536 (0.028) | 0.69 (0.022) | 16.25 (12.91) | 2863.86 (1716.43) |
| | zero | 0.4 (0.025) | 0.438 (0.023) | 32.88 (7.54) | 5851.67 (959.41) |
| | few | 0.428 (0.028) | 0.488 (0.023) | 47.99 (15.90) | 8787.62 (1120.93) |

Table 4: Performance metrics for $\zeta$ using different judge models and evaluation prompts on GPQA question/answer pairs (N=400).

| Judge Model | Prompt | Macro F1 | Coverage | Time (s) | Tokens |
|---|---|---|---|---|---|
| Claude 3.5 Sonnet | direct | 0.667 (0.021) | 1.0 (0.0) | 14.52 (2.92) | 1360.21 (386.44) |
| | zero | 0.664 (0.022) | 1.0 (0.0) | 19.29 (2.80) | 2161.70 (369.97) |
| | few | 0.66 (0.022) | 1.0 (0.0) | 23.64 (2.85) | 5055.46 (382.93) |
| Claude 3.5 Haiku | direct | 0.533 (0.023) | 1.0 (0.0) | 14.25 (2.23) | 1346.54 (377.10) |
| | zero | 0.556 (0.024) | 1.0 (0.0) | 19.70 (2.44) | 2070.33 (328.06) |
| | few | 0.577 (0.023) | 0.998 (0.002) | 19.48 (1.55) | 4829.19 (327.03) |
| Llama 4 Maverick | direct | **0.722** (0.021) | **1.0** (0.0) | 26.83 (21.83) | 2919.78 (825.91) |
| | zero | 0.694 (0.02) | 0.998 (0.002) | 25.20 (10.44) | 3462.93 (665.47) |
| | few | 0.717 (0.021) | 0.99 (0.005) | 24.49 (11.06) | 5756.76 (602.70) |
| Llama 4 Scout | direct | 0.636 (0.023) | 1.0 (0.0) | 30.07 (19.93) | 2467.57 (1096.58) |
| | zero | 0.577 (0.024) | 0.998 (0.002) | 24.74 (18.10) | 2689.86 (1180.22) |
| | few | 0.568 (0.023) | 1.0 (0.0) | 24.38 (18.08) | 4955.30 (1124.60) |
| GPT-4o | direct | 0.572 (0.023) | 1.0 (0.0) | **2.92** (7.85) | **546.73** (276.40) |
| | zero | 0.497 (0.024) | 1.0 (0.0) | 3.89 (4.92) | 1059.73 (276.40) |
| | few | 0.484 (0.024) | 1.0 (0.0) | 8.00 (8.50) | 3446.44 (343.78) |
| GPT-4o Mini | direct | 0.543 (0.023) | 1.0 (0.0) | 9.52 (6.61) | 1543.24 (848.31) |
| | zero | 0.444 (0.021) | 1.0 (0.0) | 6.60 (6.96) | 1722.51 (945.13) |
| | few | 0.538 (0.024) | 1.0 (0.0) | 11.41 (2.60) | 4949.93 (477.92) |

Table 5: Performance metrics for unilateral factuality evaluation using different judge models and evaluation prompts on GPQA question/answer pairs (N=400).

| Judge Model | Prompt | Macro F1 | Coverage | Time (s) | Tokens |
|---|---|---|---|---|---|
| Claude 3.5 Sonnet | direct | 0.81 (0.025) | 0.502 (0.024) | 19.22 (4.49) | 1190.43 (174.00) |
| | zero | 0.733 (0.027) | 0.615 (0.022) | 43.35 (6.44) | 3444.22 (171.12) |
| | few | 0.654 (0.031) | 0.65 (0.022) | 43.19 (5.45) | 6704.68 (161.86) |
| Claude 3.5 Haiku | direct | 0.667 (0.033) | 0.458 (0.024) | 15.81 (3.07) | 1014.98 (149.24) |
| | zero | 0.673 (0.036) | 0.385 (0.022) | 38.92 (3.14) | 3095.57 (131.72) |
| | few | 0.653 (0.033) | 0.45 (0.023) | 40.01 (4.09) | 6435.11 (158.40) |
| Llama 4 Maverick | direct | 0.673 (0.026) | **0.768** (0.02) | 21.84 (13.56) | 2008.57 (754.99) |
| | zero | 0.746 (0.029) | 0.528 (0.024) | 36.17 (10.85) | 4421.40 (448.00) |
| | few | 0.692 (0.026) | 0.715 (0.021) | 41.29 (31.55) | 6665.58 (512.60) |
| Llama 4 Scout | direct | 0.58 (0.031) | 0.572 (0.023) | 17.02 (9.05) | 1392.07 (434.73) |
| | zero | 0.677 (0.038) | 0.358 (0.022) | 43.41 (16.12) | 4049.26 (380.54) |
| | few | 0.558 (0.031) | 0.632 (0.023) | 36.90 (27.20) | 6267.85 (437.70) |
| GPT-4o | direct | 0.67 (0.026) | 0.74 (0.021) | 5.11 (5.49) | **477.22** (60.34) |
| | zero | 0.738 (0.032) | 0.44 (0.024) | 22.18 (3.49) | 3720.07 (314.42) |
| | few | **0.833** (0.026) | 0.482 (0.024) | 21.00 (4.86) | 6079.94 (292.95) |
| GPT-4o Mini | direct | 0.604 (0.027) | 0.718 (0.021) | **2.53** (0.72) | 483.86 (68.15) |
| | zero | 0.525 (0.038) | 0.378 (0.023) | 23.75 (10.23) | 3812.08 (829.15) |
| | few | 0.586 (0.034) | 0.472 (0.023) | 30.69 (16.02) | 6298.49 (254.71) |

Table 6: Performance metrics for $\zeta$ using different judge models and evaluation prompts on SimpleQA question/answer pairs (N=400).

| Judge Model | Prompt | Macro F1 | Coverage | Time (s) | Tokens |
|---|---|---|---|---|---|
| Claude 3.5 Sonnet | direct | **0.705** (0.022) | **1.0** (0.0) | 6.34 (1.59) | 453.53 (80.98) |
| | zero | 0.682 (0.022) | 1.0 (0.0) | 16.28 (5.05) | 1499.60 (112.18) |
| | few | 0.661 (0.023) | 1.0 (0.0) | 16.82 (1.99) | 4331.51 (89.63) |
| Claude 3.5 Haiku | direct | 0.595 (0.023) | 1.0 (0.0) | 6.69 (1.60) | 489.73 (91.76) |
| | zero | 0.55 (0.025) | 1.0 (0.0) | 16.29 (3.87) | 1505.90 (93.60) |
| | few | 0.523 (0.024) | 1.0 (0.0) | 17.02 (1.55) | 4332.01 (63.46) |
| Llama 4 Maverick | direct | 0.643 (0.023) | 1.0 (0.0) | 6.71 (4.20) | 814.69 (350.94) |
| | zero | 0.663 (0.023) | 0.992 (0.004) | 14.88 (8.78) | 2097.99 (230.58) |
| | few | 0.648 (0.023) | 0.992 (0.004) | 16.97 (10.19) | 4524.26 (265.16) |
| Llama 4 Scout | direct | 0.578 (0.023) | 1.0 (0.0) | 5.73 (4.96) | 511.43 (286.82) |
| | zero | 0.559 (0.025) | 1.0 (0.0) | 8.62 (8.44) | 1192.79 (516.58) |
| | few | 0.552 (0.024) | 1.0 (0.0) | 5.08 (6.68) | 3306.05 (376.05) |
| GPT-4o | direct | 0.62 (0.023) | 1.0 (0.0) | 2.55 (5.42) | **220.40** (30.07) |
| | zero | 0.614 (0.024) | 1.0 (0.0) | 3.49 (3.70) | 733.39 (30.07) |
| | few | 0.632 (0.023) | 1.0 (0.0) | 7.38 (9.60) | 3088.39 (30.07) |
| GPT-4o Mini | direct | 0.587 (0.023) | 1.0 (0.0) | **1.08** (0.19) | 220.58 (30.79) |
| | zero | 0.528 (0.023) | 1.0 (0.0) | 1.56 (1.52) | 788.69 (185.84) |
| | few | 0.572 (0.022) | 1.0 (0.0) | 6.75 (2.37) | 3923.80 (318.41) |

Table 7: Performance metrics for unilateral factuality evaluation using different judge models and evaluation prompts on SimpleQA question/answer pairs (N=400).

## D.4. Truth value distributions

| Judge Model | Prompt | $\langle t, t \rangle$ | $\langle t, f \rangle$ | $\langle f, t \rangle$ | $\langle f, f \rangle$ |
|---|---|---|---|---|---|
| Claude 3.5 Sonnet | direct | 0.202 (0.018) | 0.392 (0.022) | 0.355 (0.022) | 0.05 (0.011) |
| | zero | 0.435 (0.023) | 0.218 (0.02) | 0.325 (0.021) | 0.022 (0.007) |
| | few | 0.448 (0.023) | 0.18 (0.018) | 0.36 (0.022) | 0.012 (0.005) |
| Claude 3.5 Haiku | direct | 0.11 (0.015) | 0.53 (0.023) | 0.248 (0.02) | 0.112 (0.015) |
| | zero | 0.53 (0.023) | 0.208 (0.019) | 0.205 (0.019) | 0.058 (0.011) |
| | few | 0.502 (0.024) | 0.212 (0.02) | 0.225 (0.02) | 0.06 (0.011) |
| Llama 4 Maverick | direct | 0.04 (0.009) | 0.442 (0.023) | 0.41 (0.023) | 0.108 (0.015) |
| | zero | 0.37 (0.022) | 0.245 (0.02) | 0.372 (0.023) | 0.012 (0.005) |
| | few | 0.155 (0.017) | 0.438 (0.023) | 0.368 (0.022) | 0.04 (0.009) |
| Llama 4 Scout | direct | 0.072 (0.013) | 0.368 (0.023) | 0.345 (0.022) | 0.215 (0.019) |
| | zero | 0.525 (0.024) | 0.245 (0.021) | 0.215 (0.019) | 0.015 (0.006) |
| | few | 0.33 (0.022) | 0.385 (0.024) | 0.258 (0.02) | 0.028 (0.007) |
| GPT-4o | direct | 0.082 (0.013) | 0.358 (0.022) | 0.348 (0.023) | 0.212 (0.018) |
| | zero | 0.502 (0.023) | 0.108 (0.015) | 0.372 (0.022) | 0.018 (0.006) |
| | few | 0.468 (0.024) | 0.155 (0.017) | 0.362 (0.022) | 0.015 (0.006) |
| GPT-4o Mini | direct | 0.172 (0.018) | 0.145 (0.017) | 0.545 (0.023) | 0.138 (0.016) |
| | zero | 0.555 (0.023) | 0.018 (0.006) | 0.42 (0.023) | 0.008 (0.004) |
| | few | 0.51 (0.023) | 0.028 (0.007) | 0.46 (0.022) | 0.002 (0.002) |

Table 8: Truth value probabilities for $\zeta$ using different judge models and evaluation prompts for GPQA.

| Judge Model | Prompt | $\langle t, t \rangle$ | $\langle t, f \rangle$ | $\langle f, t \rangle$ | $\langle f, f \rangle$ |
|---|---|---|---|---|---|
| Claude 3.5 Sonnet | direct | 0.052 (0.01) | 0.218 (0.02) | 0.285 (0.021) | 0.445 (0.024) |
| | zero | 0.245 (0.02) | 0.152 (0.016) | 0.462 (0.022) | 0.14 (0.016) |
| | few | 0.278 (0.021) | 0.118 (0.014) | 0.532 (0.023) | 0.072 (0.012) |
| Claude 3.5 Haiku | direct | 0.035 (0.008) | 0.282 (0.022) | 0.175 (0.018) | 0.507 (0.024) |
| | zero | 0.148 (0.016) | 0.145 (0.016) | 0.24 (0.02) | 0.468 (0.023) |
| | few | 0.132 (0.016) | 0.162 (0.017) | 0.288 (0.021) | 0.418 (0.023) |
| Llama 4 Maverick | direct | 0.088 (0.013) | 0.588 (0.022) | 0.18 (0.018) | 0.145 (0.016) |
| | zero | 0.438 (0.024) | 0.35 (0.022) | 0.178 (0.018) | 0.032 (0.008) |
| | few | 0.218 (0.019) | 0.525 (0.023) | 0.19 (0.019) | 0.068 (0.012) |
| Llama 4 Scout | direct | 0.125 (0.015) | 0.368 (0.022) | 0.205 (0.018) | 0.302 (0.021) |
| | zero | 0.62 (0.022) | 0.245 (0.02) | 0.112 (0.015) | 0.022 (0.007) |
| | few | 0.232 (0.02) | 0.53 (0.023) | 0.102 (0.014) | 0.132 (0.016) |
| GPT-4o | direct | 0.058 (0.011) | 0.462 (0.023) | 0.278 (0.022) | 0.202 (0.019) |
| | zero | 0.56 (0.024) | 0.105 (0.014) | 0.335 (0.023) | 0.0 (0.0) |
| | few | 0.51 (0.024) | 0.205 (0.018) | 0.278 (0.021) | 0.008 (0.004) |
| GPT-4o Mini | direct | 0.145 (0.017) | 0.228 (0.019) | 0.49 (0.023) | 0.138 (0.016) |
| | zero | 0.62 (0.023) | 0.055 (0.01) | 0.322 (0.022) | 0.002 (0.002) |
| | few | 0.488 (0.022) | 0.272 (0.02) | 0.2 (0.018) | 0.038 (0.009) |

Table 9: Truth value probabilities for $\zeta$ using different judge models and evaluation prompts for SimpleQA.

