# OpenReview forum: "Sound and Complete Neurosymbolic Reasoning with LLM-Grounded Interpretations"
_nesyconf.org/NeSy/2025/Conference_Phase_2 — NeSy 2025 - Phase 2 Poster_

### Official Review · Reviewer_hVsf · 2025-06-27
**Sound and Complete Neuro-symbolic Reasoning with LLM-Grounded Interpretations**

**Rating:** 3
**Confidence:** 4

**Review:**

The authors propose directly integrating an LLM into the formal semantics of paraconsistent logic as a means to leverage the LLM as a knowledge base yet cope with its inherent inconsistency and completeness by leveraging pairs of tri-valued truth values.  The authors position the idea of integrating the LLM into the semantics itself, and point out how this differs from other approaches that seek to integrate LLM's with reasoning (in Figure 2).

However, I have a lot of difficult accepting the result of an LLM, which is stochastic and prone to hallucination, as part of the semantics for several reasons.  First, the interpretation derived from the LLM will change in every session of LLM, so even if the LLM it totally deterministic, any type of soundness is with respect to that session at best.  Second, even within a given session, an LLM can produce different results due to its stochastic nature.  The authors do not account for this in the logic (and instead they state that multiple prompting techniques could be used to reduce stochasticity).  Third, even if you have temperature settings in the LLM set in a fashion where it gives a deterministic result (which is generally not recommended), the approach is contingent on the verbalization functions consistently grounding language, which is unlikely.  It is easy to create multiple equivalent formulas with very different syntax – it is hard to believe the verbalization functions (as proposed by the authors) would produce consistent results for such a set of formulas.  Fourth, with the LLM in the semantics are the issues with hallucination - why should we allow a function that hallucinates to be part of the semantics?  This is difficult to accept as any formal results proved will always have the caveat that we assume the LLM does not hallucinate.

The other item that was difficult to accept was the choice of the logic itself.  The logic itself is very deterministic, which seems to not be a good fit for a stochastic LLM.  The authors mention that the approach could work with multi-prompt techniques, but the use of multiple prompts suggested by the authors only attempts to mask the nondeterminism - when in fact you have a distribution over LLM results (this makes me think a probabilistic or annotated logic might have been a better fit - see classic work by Nilsson for the former or Kifer for the later).

Their approach was also quite straight forward.  I think very little separates what the authors explored to make it distinct from one of the many prompting techniques that have been introduced in the last few years – techniques which should have been used as baselines in the experiments.  It is hard to discern what value this gives over the other prompting techniques, or if it is even equivalent to one of them.

On the whole, I felt like this paper was more of an initial idea, perhaps suited for a workshop or some other preliminary discussion.  I think the idea of integrating an LLM into semantics needs a lot more care to be a convincing approach from a theoretical perspective.  From an empirical perspective, there needs to be a much more in-depth evaluation.

**Anonymity:**

Remain anonymous

---

### Official Review · Reviewer_gbKh · 2025-07-06
**Sound and Complete Neuro-symbolic Reasoning with LLM-Grounded Interpretations**

**Rating:** 4
**Confidence:** 3

**Review:**

The paper proposes an approach that integrates LLM with paraconsistent formal reasoning to leverage the parametric knowledge embedded in LLM models while compensating for logical inconsistencies generated by LLM in its outputs. The LLM is used an interpretation function of the formal semantics of paraconsistent logic. Specifically, LLM is used to determine a bilateral factual evaluation of a sentence, stating one of three possible values for the case of be verified, and one of three possible values for the case of be refuted. Because LLM may change the evaluation sentences over time, the ground LLM interpretation is embedded into an analytic tableau reasoning that preserves the interpretation of atomic sentence stable within the scope of a reasoning process. The paper presents some theoretical results on the preservation of soundness  and completeness of a reasoning process  built from an LLM-based ground interpretation.

Strengths:
1.	The integration of LLM and paraconsistent logic is novel.
2.	The authors provide theoretical results of their proposed method and demonstrate some practical feasibility study.

Weaknesses:
1.	Implementations of Belnap computer are not very well developed and complete implementations do not exist.
2.	As pointed out by the authors, the construction of LLM-ground interpretation requires multiple API calls making the approach potentially inefficient. It is not clear the extent to which the caching approach can help overcoming this limitation.
3.	It would have helped to demonstrate the usage of such an approach in a real-world problem.

**Anonymity:**

Remain anonymous

---

### Official Review · Reviewer_Vbki · 2025-07-07
**Interpretation-based approach for LLM-Reasoner integration**

**Rating:** 7
**Confidence:** 4

**Review:**

This paper presents a preliminary work about the use of LLMs for generating interpretations of logical statements which are then solved by a reasoner. Authors propose to use a multi-valued logic for performing a paraconsistent reasoning process where statements are both confirmed or refuted by a judge (i.e., bilateral factuality evaluation). In this approach, the judge is an LLM. The proposal includes the necessary formal definitions for ensuring a sound and complete reasoning process with the interpretations provided by the LLM. Experiments performed on two well-known QA datasets confirm that the bilateral evaluation performs better than the unilateral one but decreasing the coverage of answered questions. Authors also evaluate the impact of the LLMs by using distilled versions, which present worst results than the flagship ones.
The presented idea is interesting and novel and serve as counterpart of the so-called LRMs. However, there are some concerns with the presented work:
1. It seems the evaluation has been performed on one atom formulas (the questions). I wonder how complex logical formulas would be treated with this method. For example, a query that requires some context where identified parts need to be interpreted separately. Similarly, I wonder how multi-hop questions could be solved in this framework?
2. According to figure 2.d there is a reasoner in the process, but experiments seem being evaluated without it because only atomic formulas are used. Some complete example showing the interaction between the LLM and the reasoner would be very helpful.
3. Which is the impact of the verbalization method to the final results, are somekind of prompt engineering needed to improve the results.
4. How ambiguity  and incompleteness in the verbalization can affect to the whole process?

Minor comment: page 9, second point remove "in bilateral evaluation".

**Anonymity:**

Remain anonymous